# A Fast North-Finding Algorithm on the Moving Pedestal Based on the Technology of Extended State Observer (ESO)

**DOI:** 10.3390/s22197547

**Published:** 2022-10-05

**Authors:** Yunchao Bai, Bing Li, Haosu Zhang, Sheng Wang, Debao Yan, Ziheng Gao, Wenchao Pan

**Affiliations:** 1State Key Laboratory for Manufacturing Systems Engineerng, Xi’an Jiaotong University, Xi’an 710054, China; 2International Joint Research Laboratory for Micro/Nano Manufacturing and Measurement Technologies, Xi’an Jiaotong University, Xi’an 710049, China; 3School of Marine Engineering and Technology, Sun Yat-Sen University, Guangzhou 510275, China; 4Southern Marine Science and Engineering Guangdong Laboratory (Zhuhai), Zhuhai 519000, China; 5Huazhong Institute of Electro-Optics, Wuhan National Laboratory for Optoelectronics, Wuhan 430223, China; 6CSSC Systems Engineering Research Institute, Beijing 100094, China; 7Yunnan Tongqu Engineering Testing Co., Ltd., Kunming 650000, China

**Keywords:** inertial alignment, inertial alignment, shipborne inertial systems, fiber-optic gyroscope (FOG)

## Abstract

We propose a kind of fast and high-precision alignment algorithm based on the ESO technology. Firstly, in order to solve the problems of rapid, high-accuracy, and anti-interference alignment on the moving pedestal in the north-seeker, the ESO technology in control theory is introduced to improve the traditional Kalman fine-alignment model. This method includes two stages: the coarse alignment in the inertial frame and fine alignment based on the ESO technology. By utilizing the ESO technology, the convergence speed of the heading angle can be greatly accelerated. The advantages of this method are high-accuracy, fast-convergence, strong ability of anti-interference, and short time-cost (no need of KF recursive calculation). Then, the algorithm model, calculation process, and the setting initial-values of the filter are shown. Finally, taking the shipborne north-finder based on the FOG (fiber-optic gyroscope) as the investigated subject, the test on the moving ship is carried out. The results of first off-line simulation show that the misalignment angle of the heading angle of the proposed (traditional) method is ≤2.1′ (1.8′) after 5.5 (10) minutes of alignment. The results of second off-line simulation indicate that the misalignment angle of the heading angle of the proposed (traditional) method is ≤4.8′ (14.2′) after 5.5 (10) minutes of alignment. The simulations are based on the ship-running experimental data. The measurement precisions of Doppler velocity log (DVL) are different in these two experiments.

## 1. Introduction

North-seeking (i.e., alignment of the attitude angles) is very important for people’s life and scientific research. The north-seeker is an ideal orientation equipment for mobile launch of the strategic, tactical, and campaign weapons. It can provide azimuth reference for missiles, radars, artillery, and vehicles. The portability, automation, rapidity of north-finding, and high-precision are the development-trend of the north-seeker. With the development of inertial-navigation technology, gyro north-seeker has been widely used in the military and civil field. It does not need external information and can find north independently all day and all time. It has always been a research hotspot. Presently, it is urgently hoped that the north-finder can not only be high-precision but also be rapid (the short north-finding time).

The performance indexes (the accuracy and time of north-seeking) of the gyro north-seeker are mainly affected by the north-finding principle and calculation method. Recently, the methods of gyro north-seeking mainly include the methods of compass and rotation-velocity measurement. The north-finding accuracy of the compass method is high (generally higher than 5″ (1δ)). However, its north-seeking time is long (generally more than 10 min). Therefore, it is difficult to miniaturize, and its cost is very high. The method of rotation-velocity measurement utilizes the gyroscope to measure the horizontal north-component of the earth’s rotation-angle-velocity (ω_ie_), and the angle between the gyroscope’s main-axis and the true north direction is calculated. This article mainly studies the north-seeker based on the method of rotation-velocity measurement.

At present, the research about north-seeker includes basic inertial navigation error model [1,2], observability analysis [3,4], in-drilling alignment with general dynamic error model [5], alignment algorithms based on gyro-compassing mode [6,7], alignment based on the interacting multiple model and the Huber methods [8], rapid fine alignment under marine mooring condition [9], alignment for SINS (strapdown inertial navigation system) in vehicular environment [10], alignment based on Riccati Equation and EM (expectation-maximization) convergence [11], alignment based on adjustment on separate-bias Kalman filter [12], application of nonlinear filtering in alignment [13], initial attitude estimation of tactical grade inertial measurement [14], alignment with robust adaptive unscented Kalman filter [15], alignment based on a group of double direct spatial isometries [16], alignment with state-dependent extended Kalman filter [17], application of redundant technology in north-finding [18], two-position algorithm [19,20], multi-position algorithm [21,22], rotary-modulation algorithm [23,24,25], nonlinear filter model for large misalignment angle [26], transfer north-seeking algorithm [27,28,29], transfer alignment based on cubature Kalman filter (CKF) method [30], north-finding based on the neural network technology [31,32], transfer algorithm based on sensors network and estimation of wing flexure deformation [33], fast stationary initial alignment based on extended measurement information [34], accurate fine alignment based on adaptive extended Kalman filters [35], compact north-seeker technology [36], north-seeking based on the information fusion technology [37], and so on. Most of the theory and application research about the north-seeking algorithm of strapdown inertial navigation system (SINS) adopts the Kalman filter (KF) method. However, the weak observability of the inertial navigation system affects the convergence speed and estimation accuracy of the filter for the state estimation. Then, it impacts the accuracy and rapidity of north-seeking algorithm [23], resulting in the contradiction between the accuracy and rapidity. In order to solve this contradiction, by analyzing the error model and the factors affecting the algorithm accuracy on the moving base (e.g., ship and vehicle), a method for high-precision and quick north-seeking is proposed.

As early as 1988, some scholars had proposed that the automatic control theory can be applied in the study of north-seeker [38]. Various observers are widely used in the control field. The bias observers are used for a class of uncertain nonlinear systems [39]. The ESO is utilized in the active disturbance rejection control [40], model free predictive current control [41], incremental actor faults [42], robust predictive current control [43], heat pump system [44], non-iteration predicted entry guidance [45], active disturbance rejection control [46], path following control for underactuated vehicles [47], and shipborne transfer alignment under the uncertain disturbance [48].

In the following, some existing research work closely related to this paper are selected for detailed discussion. The literature [16] had proven that the group state model satisfies a particular “group affine” property and the corresponding error model satisfies a “log-linear” autonomous differential equation on the Lie algebra. Based on this property, the attitude can be estimated based on the linear error model with even extreme large misalignments. Hence, the coarse alignment can be omitted. The simulation results of intermediate-grade SINS indicate that the pitch, roll, and heading misalignment angles are ≤6′, ≤6′, and ≤52′ after 150 s, 150 s, and 200 s of alignments, respectively. Ref. [17] proposed a novel initial alignment method by integrating the advantages of the special orthogonal group of order 3 [SO(3)] representation. This method is called as the state-dependent extended Kalman filter (SDEKF). The experimental results of navigation-grade SINS indicate that the pitch, roll and heading misalignment angles are ≤0.9′, ≤1.0′, and ≤5.6′ after 413 s of alignment, respectively. However, the mathematical fundamentals of these two alignments [16,17] are obscure. Additionally, the models, calculation process, and deductions of these methods are complicated. The article [18] had studied the alignment of regular tetrahedron RSINS (Redundant-SINS). A 13-dimensional error state equation and the Kalman filter model of fine alignment were established. The emulation results of navigation-grade SINS indicate that the pitch, roll, and heading misalignment angles are ≤0.6′, ≤0.6′, and ≤5.8′ after 3 s, 3 s, and 150 s of alignments, respectively. However, this method requires redundant inertial sensors. Hence, the structure and fabrication of SINS is complicated, and its cost is high. The literature [19] proposed a high accuracy two-position alignment for lunar rovers. Ref. [20] has proposed a rapid SINS two-position ground alignment. A two-position ground alignment algorithm for SINS is designed based on the proposed piecewise combined Kalman filter. The calculation results of navigation-grade SINS indicate that the heading misalignment angle is ≤0.5′ after 300 s of north-finding. A kind of multi-position alignment algorithm was studied in refs. [21,22]. The advantages of this kind of algorithm are high-accuracy and short time-cost. Especially, the performance of multi-position alignment algorithm is better than two-position alignment algorithm. However, such algorithms all require a precise rotating mechanism, some need the assistance of star sensor [19], and some are only suitable for static base [20]. Therefore, these algorithms are not suitable for the portable shipborne north-finder. Ref. [23] proposed a fast initial alignment based on the error model of the SINS. The algorithm is high accuracy, but it can only be applied to static base. Refs. [24,25] had proposed an alignment method based on the technology of continuous-rotation modulation. The performance of this method is better than the multi-position alignment. However, this method also requires a precise rotating mechanism, which will greatly increase the weight and volume of the north-finder. The literature [26] has introduced nonlinear-filtering technology into alignment, which has improved accuracy and shortened convergence time. However, the calculation process of this method is complex and requires a chip or hardware platform with high performance. This will increase the cost of the north-finder. Refs. [27,28,30], respectively, proposed the applications of KF, EKF (extended KF), and CKF to transfer alignment. This kind of algorithm has very high accuracy and extremely short time-cost. However, it needs a high-precision navigation coordinate system (such as north-east-up frame) as the reference benchmark. Therefore, it is not suitable for north-finder, which acquires autonomous north-seeking. The literature [31] proposed the initial alignment based on wavelet neural network. This algorithm is very advanced and has excellent performance. However, compared with the traditional KF alignment algorithm, its calculation process is much more complex. Ref. [33] proposed a transfer alignment based on inertial sensors network. The accuracy of this transfer algorithm is very high. However, because it needs a network of sensors, the cost is very high when it is applied in engineering practices. Ref. [34] proposed a fast initial alignment based on extended measurement information. This method introduces many new observations and greatly improves the alignment accuracy. However, it can only be applied to static pedestals. The literature [35] has studied a fine alignment based on adaptive extended KF. This method negates the need for multi-position rotation or a complex rotating mechanism. The fine alignment process is aimed to reduce initial attitude and inertial sensor errors before system operation. Results show that a proper choice of an adaptive filter in the alignment process can greatly improve the accuracy of the navigation error-states. The numerical-simulation results of MEMS (micro-electromechanical systems) SINS indicate that the pitch and roll misalignment angles are ≤78′ and ≤79′ after 2 s of alignments, respectively. This method not only can provide the fast and accurate estimation of the alignment error but also can predict the gyro bias online. However, the establishment and improvement of the extended measurement equation is complicated. Ref. [48] studied a shipborne transfer alignment based on ESO. This study verifies that ESO can improve the accuracy and anti-interference ability of transfer alignment. A method has been well designed to improve the Kalman filter techniques based on the extended state observer. The usage of the method makes it possible to estimate the disturbance of the uncertain model and realizes dynamic feedback linearization in the inertial navigation system for ship during the process of transfer alignment. The emulation results of navigation-grade SINS indicate that the pitch, roll, and heading misalignment angles are ≤2′, ≤2′, and ≤3′ after 10 s, 10 s, and 15 s of alignments, respectively. However, the application of transfer alignment is limited, compared with self-alignment. The study of this paper should be expanded and deepened.

In order to make up for the above deficiencies, in this article, the ESO and tracking differentiator (TD) are used to the north-finding algorithm on the motive base. The innovations of this paper include: ① In order to solve the problems of slow convergence, large amount of calculations, and poor anti-interference ability of KF fine alignment, the proposed method introduces ESO into the autonomous alignment and quickly obtains more accurate east and north misalignment angles. ② The differential of the east misalignment angle is required to calculate the azimuth misalignment angle. However, it is difficult to obtain this differential value. Here, TD filter is used to calculate the differential value of the east misalignment angle. Finally, the azimuth misalignment angle is obtained from this differential value and the north misalignment angle. ③ For faster convergence, the parameters of ESO and TD filters are determined by optimization software through a large number of simulations. This method promotes the convergence speed of all the pitch, roll, and heading misalignment angles. The advantages of this method are high accuracy, fast convergence, strong ability of anti-interference, and short time-cost (only ESO filter and TD filter equations need to be solved, and KF recursive calculation is not required). The disadvantage of this method is that only three misalignment angles can be obtained, while the KF alignment method can estimate a set of navigation parameters. If other navigation parameters (such as position, velocity, and so on) are needed after fine alignment for calculation of integrated navigation, the KF is indispensable.

The organization of this paper is as follows: firstly, the method of coarse alignment is introduced. Then, the KF model of fine alignment is established, and the KF model is simplified according to the motion characteristics of the ship, and the expressions of ESO filter and TD filter are derived. Next, the algorithm is tested on a static base. Then, the accuracy, convergence speed, and anti-interference ability of the algorithm are verified by experiments of ship running. Finally, a summary is given.

The innovations of this method are summarized as follows: (1) The ESO filter is introduced to replace the traditional KF fine alignment algorithm to obtain the roll and pitch angles. (2) Combined with the results of ESO, the TD filter is utilized to achieve accurate and fast alignment of heading angle. (3) The parameters of ESO and TD filters are adjusted to be a set of optimized values. The advantages of the proposed algorithm are: ① The convergence of heading angle is fast. ② The alignment precision is high. ③ The ability of anti-interference is relatively strong. ④ The consumed computation time is short, and the required hardware resources are very few.

## 2. Theory

As mentioned above, this method includes two stages: ① Coarse alignment in the inertial frame. ② Fine alignment based on the ESO technology. They are explained as follows.

### 2.1. Coarse Alignment in Inertial Frame

Two important inertial coordinate systems are defined: ① the inertial system (*b*_0_) of carrier at the initial time. It coincides with the carrier coordinate system (*b*) at the beginning of the north-seeking. Subsequently, no rotation of *b*_0_ frame relative to the inertial space exists; ② the navigation inertial system (*n*_0_) at the initial time. It coincides with the navigation coordinate system (i.e., the geographic coordinate system, *n* frame) at the beginning of the north-seeking. Later, no rotation of *n* frame relative to the inertial space occurs. The key to indirect initial alignment method is to solve the azimuth relationship between systems, that is, constant matrix $C_{b_{0}}^{n_{0}}$. In this article, the variable representing vector or matrix is written in bold.

Referring to Figure 1, one can observe the gravity vector ($g_{0}^{n_{0}}$) of a fixed point on the earth’s surface in the inertial coordinate system. Then, the direction of the vector will gradually change with the rotation of earth. It will rotate exactly once within 24 h to form a cone. The direction of $g_{0}^{n_{0}}$ is the sky-ground direction. Additionally, the direction of change rate (${\dot{g}}_{0}^{n_{0}}$) of $g_{0}^{n_{0}}$ is the east-west direction. Therefore, the horizontal and azimuth information of geographic coordinate system is included in the $g_{0}^{n_{0}}$ and ${\dot{g}}_{0}^{n_{0}}$. In Figure 1, $g_{0}^{n_{0}}$, $g_{t}^{n_{0}}$ and $\omega_{ie}^{}$ are the $g_{}^{n_{0}}$ at the initial time (0 time), $g_{}^{n_{0}}$ at the *t* time (0 time) and the rotation rate of the earth. The $g_{}^{n_{0}}$ is the projection of the gravity vector g in the *n*_0_ coordinate system.

Firstly, the projection of g in the *n*_0_ frame is:(1)$$g^{n_{0}}=C_{n}^{n_{0}}g^{n}$$
where $g^{n}={\left[\begin{array}{ccc} 0 & 0 & -g \end{array}\right]}^{T}$ is a constant vector, and $C_{n}^{n_{0}}$ is attitude transition matrix from *n* frame to *n**_0_* frame. The equation about $C_{n}^{n_{0}}$ can be obtained:(2)$${\dot{C}}_{n}^{n_{0}}=C_{n}^{n_{0}}(\omega_{n_{0}n}^{n}\times )=C_{n}^{n_{0}}(\omega_{ie}^{n}\times )$$

Since $\omega_{ie}^{n}=\left[0\;\omega_{ie}^{}\cos L\;\omega_{ie}^{}\sin L\right]$ is a constant value, the rotation ($\omega_{n_{0}n}^{}$) of *n* frame relative to *n*_0_ frame is around a fixed axis. $\omega_{n_{0}n}^{n}$ is the rotation of *n* frame relative to *n*_0_ frame projected in the *n* frame. Here, *ω_ie_* is the rotation rate of the earth, and *L* is the latitude. According to Equation (2), we can obtain:(3)$$\begin{array}{ll} C_{n}^{n_{0}} & =e^{\left(t\omega_{ie}^{n}\times \right)}=I+\frac{\sin\omega_{ie}t}{\omega_{ie}t}(t\omega_{ie}^{n}\times )+\frac{1-\cos\omega_{ie}t}{{\left(\omega_{ie}t\right)}^{2}}{(t\omega_{ie}^{n}\times )}^{2}\\ & =\left[\begin{array}{ccc} \cos\omega_{ie}t & -\sin\omega_{ie}t\sin L & \sin\omega_{ie}t\cos L\\ \sin\omega_{ie}t\sin L & 1-(1-\cos\omega_{ie}t)\sin^{2}L & (1-\cos\omega_{ie}t)\sin L\cos L\\ -\sin\omega_{ie}t\cos L & (1-\cos\omega_{ie}t)\sin L\cos L & 1-(1-\cos\omega_{ie}t)\cos^{2}L \end{array}\right] \end{array}$$

Hence,
(4)$$g^{n_{0}}=-g\left[\begin{array}{c} \sin\omega_{ie}t\cos L\\ (1-\cos\omega_{ie}t)\sin L\cos L\\ 1-(1-\cos\omega_{ie}t)\cos^{2}L \end{array}\right]$$

Here, *t* is the time. Secondly, the specific force output ($f_{\mathrm{sf}}^{b_{0}}$) of the accelerometer is projected in the *b*_0_ frame as:(5)$$f_{\mathrm{sf}}^{b_{0}}=C_{b}^{b_{0}}f_{\mathrm{sf}}^{b}$$
where $f_{\mathrm{sf}}^{b}$ is the specific force output of the accelerometer projected in the *b* frame. $C_{b}^{b_{0}}$ is attitude transition matrix from *b* frame to *b**_0_* frame. The equation about $C_{b}^{b_{0}}$ can be obtained where
(6)$${\dot{C}}_{b}^{b_{0}}=C_{b}^{b_{0}}(\omega_{b_{0}b}^{b}\times )=C_{b}^{b_{0}}(\omega_{ib}^{b}\times )$$

Here, $\omega_{ib}^{b}$ is the measured value of the gyroscope. The initial value ($C_{b}^{b_{0}}(0)$) of the attitude matrix is *I* (identity matrix). $\omega_{b_{0}b}^{b}$ is the rotation of *b* frame relative to *b*_0_ frame projected in the *b* frame. The real-time attitude matrix ($C_{b}^{b_{0}}$) can be obtained by using the algorithm of attitude update. There is no need to limit the magnitude of $\omega_{ib}^{b}$. Therefore, the ability of anti-angular-shaking interference of this coarse north-seeking algorithm is strong.

Finally, the relationship between the gravity acceleration and measurement of specific force of accelerometer is established by $C_{b_{0}}^{n_{0}}$. These results are obtained:(7)$$C_{n}^{n_{0}}(C_{b}^{n}{\tilde{f}}_{\mathrm{sf}}^{b}-{\overset{⌢}{\nabla }}^{n})=C_{n}^{n_{0}}(-g^{n})$$

That is:
(8)$$C_{b_{0}}^{n_{0}}(C_{b}^{b_{0}}{\tilde{f}}_{\mathrm{sf}}^{b}-{\overset{⌢}{\nabla }}^{b_{0}})=-g^{n_{0}}$$
where ${\overset{⌢}{\nabla }}^{b_{0}}=C_{b}^{b_{0}}\delta f_{\mathrm{sf}}^{b}+{\dot{v}}^{b_{0}}$ refers to the measurement error ($\delta f_{\mathrm{sf}}^{b_{0}}$) of the accelerometer and the interference of linear acceleration (${\dot{v}}^{b_{0}}$) in the *b*_0_ frame. The ${\tilde{f}}_{\mathrm{sf}}^{b}$ is defined as ${\tilde{f}}_{\mathrm{sf}}^{b}=f_{\mathrm{sf}}^{b}+\delta f_{\mathrm{sf}}^{b}$.

By Equation (8), theoretically, the two measured values of ***g*** and $f_{\mathrm{sf}}^{b}$ at two different times are obtained, and two matrix equations are established; $C_{b_{0}}^{n_{0}}$ can be solved by the algorithm of attitude determination based on the double vectors. However, in order to reduce the influence of the interference of linear motion, we can integrate Equation (8) during the whole coarse north-seeking and obtain:(9)$$\left\{\begin{array}{l} {\tilde{F}}_{i}^{b_{0}}=\int_{0}^{t_{i}}C_{b}^{b_{0}}{\tilde{f}}_{\mathrm{sf}}^{b}dt\\ G_{i}^{n_{0}}=-\int_{0}^{t_{i}}g^{n_{0}}dt \end{array}\right.\left(i=1,2\right)$$
where ${\tilde{F}}_{i}^{b_{0}}$ and $G_{i}^{n_{0}}$ are integrated specific force (including the bias of the accelerometer) and gravity acceleration projected in the *n*_0_ frame.

This result is acquired:(10)$$C_{b_{0}}^{n_{0}}{\tilde{F}}_{i}^{b_{0}}-\int_{0}^{t_{i}}C_{b_{0}}^{n_{0}}{\overset{⌢}{\nabla }}^{b_{0}}dt=G_{i}^{n_{0}}$$

Usually, *t*_2_ = 2*t*_1_ and *t*_2_ are taken as the end time of the coarse north-seeking.

According to the algorithm of attitude determination based on the double vectors, by ignoring the influence of ${\overset{⌢}{\nabla }}^{b_{0}}$ in Equation (9), Equation (10) can be obtained:(11)$${\hat{C}}_{b_{0}}^{n_{0}}=\left[\frac{G_{1}^{n_{0}}}{\left|G_{1}^{n_{0}}\right|}\;\frac{G_{1}^{n_{0}}\times G_{2}^{n_{0}}}{\left|G_{1}^{n_{0}}\times G_{2}^{n_{0}}\right|}\;\frac{G_{1}^{n_{0}}\times G_{2}^{n_{0}}\times G_{1}^{n_{0}}}{\left|G_{1}^{n_{0}}\times G_{2}^{n_{0}}\times G_{1}^{n_{0}}\right|}\right]\left[\begin{array}{c} {\left({\tilde{F}}_{1}^{b_{0}}\right)}^{T}/\left|{\tilde{F}}_{1}^{b_{0}}\right|\\ {\left({\tilde{F}}_{1}^{b_{0}}\times {\tilde{F}}_{2}^{b_{0}}\right)}^{T}/\left|{\tilde{F}}_{1}^{b_{0}}\times {\tilde{F}}_{2}^{b_{0}}\right|\\ {\left({\tilde{F}}_{1}^{b_{0}}\times {\tilde{F}}_{2}^{b_{0}}\times {\tilde{F}}_{1}^{b_{0}}\right)}^{T}/\left|{\tilde{F}}_{1}^{b_{0}}\times {\tilde{F}}_{2}^{b_{0}}\times {\tilde{F}}_{1}^{b_{0}}\right| \end{array}\right]$$

Here, ${\hat{C}}_{b_{0}}^{n_{0}}$ represents the calculated attitude matrix at the initial time of rough north-seeking. In order to obtain the calculated attitude matrix ${\hat{C}}_{b}^{n}$ at the end time of rough north-seeking, the following chain multiplication formula can be utilized:(12)$${\hat{C}}_{b}^{n}=C_{n_{0}}^{n}{\hat{C}}_{b_{0}}^{n_{0}}C_{b}^{b_{0}}$$
where $C_{n_{0}}^{n}$ and $C_{b}^{b_{0}}$ are ideal attitude matrix, and they are determined by Equations (3) and (6), respectively.

Compared with analytical rough alignment, the ability of anti-angular-shaking interference of this alignment method is better. However, the ability of anti linear-shaking interference of this method is relatively weak during a short time. If the coarse alignment time is properly extended (generally in the order of minutes), then this method can often achieve good results. This method for coarse north-finding is very practical, and it has been widely applied in many north-seekers.

### 2.2. Fine Alignment Algorithm

Before introducing the ESO method, we bring in the traditional fine alignment algorithm firstly.

The relationship between *V*_N_ (*V*_E_) and $\dot{L}$ ($\dot{\lambda }$) is expressed as Equation (13):(13)$$\left\{\begin{array}{l} \dot{L}=V_{N}/R\\ \dot{\lambda }=V_{E}\sec L/R \end{array}\right.$$

Here, *V*_N_, *V*_E_, *λ,* and *R* are northward velocity of the carrier, eastward velocity, longitude, and earth radius, respectively. $\dot{L}$ and $\dot{\lambda }$ are differentials of L and λ.

The error model of north-seeker (or strapdown inertial navigation) under the condition of moving base is established as follows:(14)$$\left[\begin{array}{c} \delta {\dot{V}}_{E}\\ \delta {\dot{V}}_{N}\\ {\dot{\phi }}_{E}\\ {\dot{\phi }}_{N}\\ {\dot{\phi }}_{U} \end{array}\right]=\left[\begin{array}{ccccc} \frac{V_{N}}{R}tgL & (2\Omega +\dot{\lambda })SL & 0 & -f_{U}^{n} & f_{N}^{n}\\ -2(\Omega +\dot{\lambda })SL & 0 & f_{U}^{n} & 0 & -f_{E}^{n}\\ 0 & -\frac{1}{R} & 0 & (\Omega +\dot{\lambda })SL & -(\Omega +\dot{\lambda })CL\\ \frac{1}{R} & 0 & -(\Omega +\dot{\lambda })SL & 0 & -\dot{L}\\ \frac{tgL}{R} & 0 & (\Omega +\dot{\lambda })CL & \dot{L} & 0 \end{array}\right]\left[\begin{array}{c} \delta {\dot{V}}_{E}\\ \delta {\dot{V}}_{N}\\ \phi_{E}\\ \phi_{N}\\ \phi_{U} \end{array}\right]+\left[\begin{array}{c} \nabla_{aE}\\ \nabla_{aN}\\ \varepsilon_{gE}\\ \varepsilon_{gN}\\ \varepsilon_{gU} \end{array}\right]$$
where Ω = *ω*_ie_, *SL* = sin*L*, and *CL* = cos*L*. The $\nabla_{aE},\nabla_{aN},\varepsilon_{aE},\varepsilon_{aN},\varepsilon_{aU}$ are bias (or drift) in the *n* frame of accelerometer and gyroscope. $f_{E}^{n}$, $f_{N}^{n}$, and $f_{U}^{n}$ are three components of the specific force projected in the *n* frame.

We establish the Kalman model:(15)$$\dot{X}=AX+W$$
where $X={\left[\begin{array}{cccccccccc} \delta V_{E} & \delta V_{N} & \phi_{E} & \phi_{N} & \phi_{U} & \nabla_{E} & \nabla_{N} & \varepsilon_{E} & \varepsilon_{N} & \varepsilon_{U} \end{array}\right]}^{T}$ is the vector of system state. $W={\left[\begin{array}{cccccccccc} w_{aE} & w_{aN} & w_{gE} & w_{gN} & w_{gU} & 0 & 0 & 0 & 0 & 0 \end{array}\right]}^{T}$ is the vector of system noise. *A* is the system matrix, and $w_{gE},w_{gN},w_{gU}$ are the white noise part of gyro bias. The $w_{aE}$ and $w_{aN}$ are the white noise part of accelerometer bias. Here,
(16)$$A=\left[\begin{array}{cc} A^{\prime } & I_{5\times 5}\\ 0_{5\times 5} & 0_{5\times 5} \end{array}\right]$$
(17)$$A^{\prime }=\left[\begin{array}{ccccc} \frac{V_{N}}{R}tgL & (2\Omega +\dot{\lambda })SL & 0 & -f_{U}^{n} & f_{N}^{n}\\ -2(\Omega +\dot{\lambda })SL & 0 & f_{U}^{n} & 0 & -f_{E}^{n}\\ 0 & -\frac{1}{R} & 0 & (\Omega +\dot{\lambda })SL & -(\Omega +\dot{\lambda })CL\\ \frac{1}{R} & 0 & -(\Omega +\dot{\lambda })SL & 0 & -\dot{L}\\ \frac{tgL}{R} & 0 & (\Omega +\dot{\lambda })CL & \dot{L} & 0 \end{array}\right]$$

All variables in this formula have been described above. The observation equation is:(18)$$\left[\begin{array}{c} \delta V_{E}\\ \delta V_{N} \end{array}\right]=\left[\begin{array}{cccccccccc} 1 & 0 & 0 & 0 & 0 & 0 & 0 & 0 & 0 & 0\\ 0 & 1 & 0 & 0 & 0 & 0 & 0 & 0 & 0 & 0 \end{array}\right]X+\left[\begin{array}{c} \eta_{E}\\ \eta_{N} \end{array}\right]$$
(19)Z=HX+η

$Z={\left[\begin{array}{cc} Z_{1} & Z_{2} \end{array}\right]}^{T}={\left[\begin{array}{cc} \delta V_{E} & \delta V_{N} \end{array}\right]}^{T}={\left[\begin{array}{cc} V_{INS-E}-V_{DVL-E} & V_{INS-N}-V_{DVL-N} \end{array}\right]}^{T}$ is the observation vector; H is the measurement matrix; $\eta ={\left[\begin{array}{cc} \eta_{x}, & \eta_{y} \end{array}\right]}^{T}$ is the observation noise, which is the Gaussian white noise with zero mean. $V_{INS-E(N)}$ is the eastward (northward) velocity calculated by the inertial navigation system or north-seeker. $V_{DVL-E(N)}$ is the eastward (northward) velocity measured by the Doppler velocity log (DVL).

Equations (15) and (19) constitute the Kalman model/system. That is:(20)$$\left\{\begin{array}{l} \dot{X}=AX+W\\ Z=HX+\eta \end{array}\right.$$

It can be proved that this system is not completely observable. Among the ten state variables $\delta V_{E},\delta V_{N},\phi_{E},\phi_{N},\phi_{U},\nabla_{E},\nabla_{N},\varepsilon_{E},\varepsilon_{N},\varepsilon_{U}$, these three state variables $\nabla_{E},\nabla_{N},\varepsilon_{E}$ are unobservable. However, the Kalman model is still available. The observed variables $\delta V_{E},\delta V_{N}$ can be utilized to estimate the state variables $\phi_{E},\phi_{N},\phi_{U},\varepsilon_{N},\varepsilon_{U}$. Therefore, the north-seeking can be completed.

The equations for time update are:(21)$${\hat{X}}_{k/k-1}=\Phi_{k/k-1}{\hat{X}}_{k-1}$$
where ${\hat{X}}_{k/k-1}$, $\Phi_{k/k-1}$, and ${\hat{X}}_{k-1}$ are the state of one-step prediction, state transition matrix, and state estimation of $X_{k-1}$, respectively.
(22)$$P_{k/k-1}=\Phi_{k/k-1}P_{k-1}\Phi_{k/k-1}^{T}+Q$$
where $P_{k/k-1}$ is the mean square error matrix of state of one-step prediction; $P_{k-1}$ is the mean square error matrix of state estimation; and Q is the system noise.

The equations for measurement update are:(23)$$K_{k}^{}=P_{k/k-1}^{}H_{k}^{T}{\left(H_{k}^{}P_{k/k-1}^{}H_{k}^{T}+R_{}^{}\right)}^{-1}$$
where $K_{k}^{}$ is the filter gain, and R is the observation noise. $P_{k}^{}$ and ${\hat{X}}_{k}^{}$ can be obtained by Equations (24) and (25):(24)$$P_{k}^{}=(I-K_{k}^{}H_{k}^{})P_{k/k-1}$$
(25)$${\hat{X}}_{k}^{}={\hat{X}}_{k/k-1}^{}+K_{k}^{}(Z_{k}^{}-H_{k}^{}{\hat{X}}_{k/k-1}^{})$$

The initial conditions are set as follows:(26)$$\left\{\begin{array}{l} L_{0}=30{}^{\circ}\\ T=0.005\;s\\ \delta V_{E0}=\delta V_{N0}=0.1\;m/s\\ \phi_{E0}=\phi_{N0}=\phi_{U0}=3{}^{\circ} \end{array}\right.$$
where *L*_0_ is the initial latitude. The latitude of experimental site is close to 30° (as shown by following Section 3). Additionally, *T* is the sampling period of inertial devices or the calculation step of the algorithm. The values of $\delta V_{E0},\delta V_{N0},\phi_{E0},\phi_{N0},\phi_{U0}$ are determined by the experience and simulations.

The initial matrixes of Kalman model are:(27)$$\left\{\begin{array}{l} P_{0}=\mathrm{diag}\left[\begin{array}{cccccccccc} {\left(0.1\;m/s\right)}^{2} & {\left(0.1\;m/s\right)}^{2} & {\left(3{}^{\circ}\right)}^{2} & {\left(3{}^{\circ}\right)}^{2} & {\left(3{}^{\circ}\right)}^{2} & {\left(2\times 10^{-5}g\right)}^{2} & {\left(2\times 10^{-5}g\right)}^{2} & {\left(0.02{}^{\circ}/h\right)}^{2} & {\left(0.02{}^{\circ}/h\right)}^{2} & {\left(0.02{}^{\circ}/h\right)}^{2} \end{array}\right]\\ Q=\mathrm{diag}\left[\begin{array}{cccccccccc} {\left(1\times 10^{-5}\;g\right)}^{2} & {\left(1\times 10^{-5}\;g\right)}^{2} & {\left(0.01{}^{\circ}/h\right)}^{2} & {\left(0.01{}^{\circ}/h\right)}^{2} & {\left(0.01{}^{\circ}/h\right)}^{2} & 0 & 0 & 0 & 0 & 0 \end{array}\right]\\ R=\mathrm{diag}\left[\begin{array}{cc} {\left(0.1\;m/s\right)}^{2} & {\left(0.1\;m/s\right)}^{2} \end{array}\right] \end{array}\right.$$
where *P*_0_ is initial value of *P_k_*. *Q* is the system noise, and *R* is the observation noise.

Equation (27) is obtained by the parameters shown in Table 1 and the results exhibited by Equation (26). The initial values shown by Equation (27) are optimal for KF, and the performance of KF with these initial values is high. Hence, the comparison between KF and ESO is fair.

The above is the traditional fine north-seeking algorithm. The following describes how to modify it to obtain the ESO method.

Generally, the carrier (such as vehicle and ship) moves in a straight line at a uniform speed. Here, the rapid north-seeking is studied under the condition of constant speed and straight-line motion of the carrier.

Under this condition, the output of the accelerometer is $f_{E}^{n}=f_{N}^{n}=0$ and $f_{U}^{n}=g$. When the speed is small, the constant speed has little effect on the system matrix. For example, when $V_{x}=\sqrt{V_{E}^{2}+V_{N}^{2}}=3\;m/s$, then $V_{x}/R=4.7\times 10^{-7}\mathrm{rad}/s$, $V_{x}/R\ll \omega_{ie}$, $V_{E}/R\ll \omega_{ie}$, $V_{N}/R\ll \omega_{ie}$. The $\dot{\lambda }$ and $\dot{L}$ in the above $A^{\prime }$ matrix can be approximately 0.

Therefore, Kalman filter model under uniform linear motion is as follows. We substitute the condition $V_{E}=V_{N}=0,f_{E}^{n}=f_{N}^{n}=0;f_{U}^{n}=g$ into Equation (17) and obtain:(28)$$A^{\prime }=\left[\begin{array}{ccccc} 0 & -2\Omega_{D} & 0 & -g & 0\\ 2\Omega_{D} & 0 & g & 0 & 0\\ 0 & -1/R & 0 & -\Omega_{D} & -\Omega_{N}\\ 1/R & 0 & \Omega_{D} & 0 & 0\\ tgL/R & 0 & \Omega_{N} & 0 & 0 \end{array}\right]$$
where $\Omega_{D}=-\omega_{ie}\sin L$ and $\Omega_{N}=\omega_{ie}\cos L$ are the down and north components of $\omega_{ie}$ projected in the *n* frame.

We record the error model shown by Equation (14) as followings:(29)$$\left\{\begin{array}{l} \delta {\dot{V}}_{E}=-2\Omega_{D}\cdot \delta V_{N}-g\cdot \phi_{N}+\nabla_{E}\\ \delta {\dot{V}}_{N}=2\Omega_{D}\cdot \delta V_{E}+g\cdot \phi_{E}+\nabla_{N}\\ {\dot{\phi }}_{E}=-\frac{1}{R}\delta V_{N}-\Omega_{D}\cdot \phi_{N}-\Omega_{N}\cdot \phi_{U}+\varepsilon_{E}\\ {\dot{\phi }}_{N}=\frac{1}{R}\delta V_{E}+\Omega_{D}\cdot \phi_{E}+\varepsilon_{N}\\ {\dot{\phi }}_{U}=\frac{tgL}{R}\delta V_{E}-\Omega_{N}\cdot \phi_{E}+\varepsilon_{U} \end{array}\right.$$

Here, the white noises are ignored. Based on the first and second formulas in Equation (29), the following is acquired:(30)$$\left\{\begin{array}{l} \phi_{N}=\frac{1}{g}(-\delta {\dot{V}}_{E}+2\Omega_{D}\cdot \delta V_{N})+\frac{\nabla_{E}}{g}\\ \phi_{E}=\frac{1}{g}(\delta {\dot{V}}_{N}-2\Omega_{D}\cdot \delta V_{E})-\frac{\nabla_{N}}{g} \end{array}\right.$$

According to the conclusions of observability analysis [3], the $\nabla_{E}$ and $\nabla_{N}$ are not observable. Hence, they cannot be estimated. All the variables $\delta {\dot{V}}_{E},\delta {\dot{V}}_{N},\delta V_{E},\delta V_{N}$ are observable. The observability analysis based on singular value decomposition or eigenvalue/eigenvector theories of KF model can be found in refs. [3,4]. Based on the steady-state analysis of error model of KF [1,2], if both the $\nabla_{E}$ and $\nabla_{N}$ are set to be 0, then the steady states of $\phi_{N}$ and $\phi_{E}$ can be obtained. Hence, the steady-state errors $\delta \phi_{N}$ and $\delta \phi_{E}$ are shown as follows:(31)$$\left\{\begin{array}{l} \delta \phi_{N}=\frac{\nabla_{E}}{g}\\ \delta \phi_{E}=-\frac{\nabla_{N}}{g} \end{array}\right.$$

From the third formula in Equation (29), we can obtain:(32)$$\phi_{U}=\frac{1}{\Omega_{N}}(-\Omega_{D}\phi_{N}-{\dot{\phi }}_{E}+\varepsilon_{E}-\frac{\delta V_{N}}{R})$$

By substituting Equation (30) into Equation (32), the following can be acquired:(33)$$\phi_{U}=-\frac{1}{g\Omega_{N}}(\delta {\ddot{V}}_{N}-3\Omega_{D}\delta {\dot{V}}_{E}-2\Omega_{D}^{2}\delta V_{N})+\frac{\varepsilon_{E}}{\Omega_{N}}-\frac{\Omega_{N}}{g\Omega_{D}}\nabla_{E}-\frac{\delta V_{N}}{R\Omega_{N}}$$

All the variables $\delta {\ddot{V}}_{N},\delta {\dot{V}}_{E}$, and $\delta V_{N}$ are observable. The variables $\nabla_{E}$ and $\varepsilon_{E}$ are not observable [3,4]. For obtaining a steady-state value of $\phi_{U}$, the parameters $\nabla_{E}$ and $\varepsilon_{E}$ in Equation (33) are taken to be 0 [1,2]. Therefore, the steady-state error ($\delta \phi_{U}$) of $\phi_{U}$ can be obtained:(34)$$\delta \phi_{U}=\frac{\varepsilon_{E}}{\Omega_{N}}-\frac{\Omega_{N}}{g\Omega_{D}}\nabla_{E}$$

The Kalman filtering results of $\phi_{E}$, $\phi_{N}$, and $\phi_{U}$ show that the convergence speed of $\phi_{E}$ and $\phi_{N}$ is much faster than that of $\phi_{U}$. Therefore, we use the steady-state value of $\phi_{E}$ and $\phi_{N}$ to estimate the $\phi_{U}$. When $\phi_{E}$ and $\phi_{N}$ converge, $\phi_{U}$ will converge. In Equation (32), assuming $\varepsilon_{E}=0.02{}^{\circ}/h$ and $\delta V_{N}=0.1\;m/s$, we obtain $\varepsilon_{E}=9.6\times 10^{-8}\mathrm{rad}/s$ and $\delta V_{N}/R=1.5\times 10^{-8}\mathrm{rad}/s$. And hence, $\varepsilon_{E}-\frac{\delta V_{N}}{R}\ll -\Omega_{D}\phi_{N}-{\dot{\phi }}_{E}$ (both $\Omega_{D}\phi_{N}$ and ${\dot{\phi }}_{E}$ are about $5.0\times 10^{-6}\mathrm{rad}/s$). We can obtain:(35)$${\bar{\phi }}_{U}=\frac{1}{\Omega_{N}}(-\Omega_{D}{\bar{\phi }}_{N}-{\dot{\bar{\phi }}}_{E})$$

In this formula, ${\bar{\phi }}_{E}$ and ${\bar{\phi }}_{N}$ are the estimated values of $\phi_{E}$ and $\phi_{N}$, respectively. Therefore, the estimation speed of ${\bar{\phi }}_{U}$ (${\bar{\phi }}_{U}$ is the estimation value of $\phi_{U}$) based on ${\bar{\phi }}_{E}$ and ${\bar{\phi }}_{N}$ will be greatly accelerated. This can be achieved by designing an ESO. We construct a measurement equation:(36)$$y_{1}={\bar{\phi }}_{E}$$

Regarding the third formula in Equations (29) and (36) as a first-order system, an ESO can be designed:(37)$$\left\{\begin{array}{l} e_{}=z_{1}-y_{1}\\ {\dot{z}}_{1}=z_{2}-\beta_{1}\cdot e_{}\\ {\dot{z}}_{2}=-\beta_{2}\cdot fal(e_{},\alpha_{},\delta_{}) \end{array}\right.$$
where *fal* is a nonlinear feedback function. The *e* is the estimated error. The *β*_1_ and *β*_2_ are adjustable parameters, and the *z*_1_ and *z*_2_ are the two outputs of ESO, which are shown by Equation (38).
(38)$$\left\{\begin{array}{l} z_{1}=\phi_{E}\\ z_{2}=-\frac{1}{R}\delta V_{N}-\Omega_{D}\cdot \phi_{N}-\Omega_{N}\cdot \phi_{U}+\varepsilon_{E} \end{array}\right.$$

The ESO equations are established based on the control theory [42,46,47]. The first expression in Equation (37) defines the error. The second formula is based on the third formula in Equation (29). The error term $-\beta_{1}\cdot e_{}$ added in this formula is to accelerate the convergence speed of *z*_1_. The third formula uses the nonlinear feedback function to express the differential of *z*_2_. After solving Equation (37), *z*_1_ and *z*_2_ are obtained.

Similarly, if the fourth expression in Equations (29) and (39) are regarded as a first-order system, an ESO can also be designed, as shown in Equation (40). Additionally, the *z*_3_ and *z*_4_ are the two outputs, which are shown by Equation (41).
(39)$$y_{2}={\bar{\phi }}_{N}$$
(40)$$\left\{\begin{array}{l} e_{}=z_{3}-y_{2}\\ {\dot{z}}_{3}=z_{4}-\beta_{3}\cdot e_{}\\ {\dot{z}}_{4}=-\beta_{4}\cdot fal(e_{},\alpha_{2},\delta_{2}) \end{array}\right.$$
(41)$$\left\{\begin{array}{l} z_{3}=\phi_{N}\\ z_{4}=\frac{1}{R}\delta V_{E}+\Omega_{D}\cdot \phi_{E}+\varepsilon_{N} \end{array}\right.$$

In Equations (37) and (40), the function *fal* is:(42)$$fal(e,\alpha_{i},\delta_{i})=\left\{\begin{array}{ll} {\left|e\right|}^{\alpha_{i}}\mathrm{sign}(e), & \left|e\right|>\delta_{i}\\ \frac{e}{\delta_{i}{}^{1-\alpha_{i}}}, & \left|e\right|\leq \delta_{i} \end{array}\right.\left(i=1,2\right)$$
where the error *e* is the input variable of the *fal* function. The *δ_i_* is any small positive number and the parameter $\alpha_{i}\in (0,1)$. There are many forms of nonlinear feedback functions to be chosen [46,47], of which the most commonly used is the *fal* function shown by formula (42). This function has the advantages of simple form, less computational efforts, high accuracy, and wide application.

The ${\bar{\phi }}_{E}$ can be estimated by Equation (37) (${\bar{\phi }}_{E}{=z}_{1}$). Similarly, ${\bar{\phi }}_{N}$ can be estimated by Equation (40) (${\bar{\phi }}_{N}{=z}_{2}$). However, ${\dot{\bar{\phi }}}_{E}$ is difficult to be calculated, and an extra tracking differential (TD) filter is needed. The TD is widely applied to study the fine alignment of platform inertial navigation systems. In this article, the ESO with the filtering function is called the ESO filter, and TD with filtering function is called TD filter. The principle and calculation method of the TD filter are similar to those of ESO filter. TD filter is introduced as follows:

Assuming $x_{1}=\phi_{E},x_{2}={\dot{\phi }}_{E}$, then the third formula in Equation (29) becomes:(43)$$\left\{\begin{array}{l} {\dot{x}}_{1}=x_{2}\\ {\dot{x}}_{2}=f(x_{1},x_{2}) \end{array}\right.$$

Equation (43) is expressed in TD form:(44)$$\left\{\begin{array}{l} {\dot{x}}_{1}=x_{2}\\ {\dot{x}}_{2}=-r\cdot sign\left[x_{1}-v+\frac{\left|x_{2}\right|\cdot x_{2}}{2r}\right] \end{array}\right.$$

In the specific implementation, the discrete form is adopted:(45)$$\left\{\begin{array}{l} x_{1}(k+1)=x_{1}(k)+h\cdot x_{2}(k)\\ x_{2}(k+1)=x_{2}(k)+h\cdot fst(x_{1}(k),x_{2}(k),v(k),r,h_{1}) \end{array}\right.$$
where *x*_1_ is the tracking signal of *v* and can be obtained by TD filter; *x*_2_ is the differential signal of *v*; *h* is the sampling step size and is set to be 5 ms; *r* is the parameter that determines the tracking speed, and it is called the speed factor; *h*_1_ is the filter factor; *fst* (•) is a nonlinear function, and the expression is:(46)$$fst(x_{1},x_{2},v,r,h_{1})=-r\cdot sat(g^{\prime },\delta )$$

Here,
(47)$$\left\{\begin{array}{l} g^{\prime }=\left\{\begin{array}{l} x_{2}-sign(y)(\delta_{00}-8r\left|y\right|+\delta_{00}^{2})/2,\left|y\right|\geq \delta_{01}\\ x_{2}+y/h_{1},\left|y\right|<\delta_{01} \end{array}\right.\\ sat(x,\delta_{00})=\left\{\begin{array}{l} sign(x),\left|x\right|\geq \delta_{00}\\ x/\delta_{00},\left|x\right|<\delta_{00} \end{array}\right. \end{array}\right.$$

The parameters in Equation (47) are:(48)$$\left\{\begin{array}{l} \delta_{00}=h_{1}r\\ \delta_{01}=h_{1}\delta_{00}\\ e=x_{1}-v\\ y=e+h_{1}x_{2} \end{array}\right.$$

The results show that when *h*_1_ > *h* (generally, *h*_1_ = 2*h*), the filtering performance of TD for the signal with noise is better, and the obtained differential signal is accurate and ideal. Formulas (43)–(48) can be used to obtain ${\dot{\phi }}_{E}$ (${\dot{\phi }}_{E}=x_{2}$).

Only when ${\bar{\phi }}_{E}$ and ${\bar{\phi }}_{N}$ are close to the steady-state values are they then converted to be estimated by the ESO. Additionally, ${\bar{\phi }}_{U}$ can be obtained by Equation (28). Generally, after coarse alignment, both the $\phi_{E}$ and $\phi_{N}$ are small enough and can be considered to be the input of ESO. The parameters in two ESOs are set as follows:(49)$$\left\{\begin{array}{l} \alpha_{1}=\alpha_{2}=0.5\\ \delta_{1}=\delta_{2}=0.025\\ \beta_{1}=\beta_{3}{=1\;s}^{-1}\\ \beta_{2}=\beta_{4}=0{.2\;s}^{-2} \end{array}\right.$$

The selection of parameters in *fal* function is based on a lot of simulations. The simulation results show that the *fal* function with the parameters shown by Equation (49) is excellent performance, high accuracy, and fast convergence. The determinations of the optimal parameters in ESO and TD are essentially a problem of multi-parameter optimization, which is very difficult to accurately solve. In this paper, the module of simulated annealing of commercial optimization software “Optimus” is applied for numerical calculation, and a set of solutions as shown in Equation (49) are obtained. The numerical simulation experiments show that this set of parameters is at least a group of local optimal solutions. Additionally, the parameters shown by Equation (49) can ensure the faster convergence speeds and higher calculation accuracies of ESO and TD.

After the coarse north-seeking, the fine north-seeking based on the ESO and TD filters are a start-up to greatly accelerate the convergence of $\bar{\phi_{U}}$ and reduce the calculating time (generally from 5 min to 0.5 min). In the process of north-seeking, the controlled object is the navigation error equation. The error without estimation by the model is calculated by using the expanded state so as to realize the dynamic-feedback linearization in the process of initial alignment. The nonlinear configuration is used to form a nonlinear feedback control law to improve the control performance of the system.

As previously analyzed, the innovations of this algorithm are as follows: ① The ESO does not include KF, as shown in Equations (37) and (40). Therefore, no matrix calculation is required. ② The TD as shown in Equation (45) solves the problems of fast convergence and high accuracy of alignment of heading angle. ③ The parameters shown by Equation (49) are obtained by optimization theory and to ensure the excellent performances of ESO and TD.

## 3. Experiment and Discussion

The experimental process and detail are shown as follows.

The north-seeker based on fiber optical gyro (FOG) is exhibited in Figure 2a. The black object in the shape of a round cake is the FOG. The interfaces of test software of the north-seeker is shown in Figure 2b.

Based on the testing reports from the company supplying the gyro and the accelerometer, the bias of the inertial devices in FOG north-seeker is shown in Table 1. The constant gyro bias, random gyro bias, constant accelerometer bias, and random accelerometer bias in FOG north-seeker are about 0.02°/h, 0.01°/h, 20 μg and 10 μg, respectively.

Both the traditional Kalman-filter (KF) algorithm and proposed ESO algorithm were embedded into the FOG north-seeker. The coarse alignment algorithms of these two methods were the same. Additionally, this method is shown in Section 2.1. The fine alignment algorithms of these two methods were traditional KF and ESO, respectively. The static north-seeking experiment on the rotary-table was performance. The misalignment angles were exhibited in Table 2.

After 5.5 min, the $\phi_{U}$ of ESO method was very small, and the convergence time of $\phi_{U}$ of traditional KF method is long. The $\phi_{E}$ and $\phi_{N}$ were easy to converge, and the differences in results of these two methods are infinitesimal. The alignment precision of this north-seeker was high because of the high performance of the inertial devices.

However, the application of static north-seeking is very limited. The dynamic north-seeking is acquired in the field exploration, mobile-launching weapon, and so on. The application value of north-seeking, orientation keeping, and course maintenance on the moving base are greater than those of static north-seeking. In order to meet the demand of dynamic orientation and positioning, the ship-running tests were carried out. In this experiment, the accurate attitude reference was very important. The high-precision north-seeker based on the exact ring laser gyro (RLG) were selected as the attitude reference. This north-seeker is shown in Figure 3.

The parameters of the sensors in RLG north-seeker were shown by Table 3, based on the factory report. Based on this table, the constant gyro bias, random gyro bias, constant accelerometer bias, and random accelerometer bias in SLG north-seeker were about 0.001°/h, 0.0005°/h, 10 μg, and 5 μg, respectively. Hence, the attitude error was very small and can be ignored. The test software of the RLG north-seeker was the same as that of FOG north-seeker.

A ship-running experiment was carried out in the Zhanghe Reservoir in Jingmen City, Hubei Province, China. The experimental ship, GNSS installed at the stern, the electronic box of DVL, and the transducer of DVL are shown in Figure 4a–d.

The proposed ESO algorithm was embedded into the ring laser gyroscope (RLG) north-seeker, and the alignment is completed under the state of mooring. After the alignment, the ship was started. Twenty minutes after the moment when RLG north-seeker completed ESO, fine alignment was considered as the starting point of the whole experiment. After that starting point, the speed of ship was stable. Therefore, the results of the attitude angles after the fine alignment can be regarded as the accurate values or benchmark values. The misalignment angles of FOG north-seeker can be calculate based on these accurate attitude angles. The whole duration of the test was about 4.8493 × 10^4^ s (about 13.4703 h). In this experiment, FOG north-seeker accomplished several times of alignment experiment. The representative two among these experiments were selected as the first and second experiment. The duration time of the first (second) experiment was 4301.4–4601.4 s (5825.7–6125.7 s). The length of duration was 10 min (5 min coarse alignment +5 min traditional KF fine alignment).

Figure 5a,b shows the sailing trajectories of the ship in the first and second experiment, respectively. These two figures were obtained based on the GNSS measurements of position. The trajectory of the ship in the first (second) experiment was straight line (curve). In order to improve the integrated alignment and navigation accuracy, both velocity measurement of DVL and position measurement of GNSS were utilized as the observation vector in the integrated algorithm of RLG north-finder. For the universality of the ESO algorithm, only the velocity measurement of DVL was applied as the observation vector in the integrated algorithms (both traditional and ESO algorithms) of FOG north-seeker.

Figure 5c,d exhibits the data of *z*-axis gyro and *z*-axis accelerometer in the first test. The average values of Figure 5c,d were 0 rad/s and 9.8 m/s^2^. Additionally, these indicated that the attitude and speed of the ship are relatively steady. Figure 5e,f exhibits the data of *z*-axis gyro and *z*-axis accelerometer in the second test. The mean value of Figure 5e was much larger than 0 rad/s during the process of turnaround. Additionally, the average value of Figure 5f was slightly larger than 9.8 m/s^2^. These imply that the attitude and speed of the ship were not steady. Due to the measurement noise of inertial devices, too dense sampling, vibration, and angular shaking of moving ships, the curves in Figure 5c–f are very thick and not clear enough. However, the mean or center line of these corpulent curves can still reflect the motion characteristics of the ship and measurement results from inertial devices.

Figure 6a,b shows the curves of the ship speed obtained by the DVL measurement in the first and second test. Turning a corner implied slowing down and caused unstable attitude of the ship and impacted the velocity measurement of DVL. Therefore, the measurement precision of DVL was high/low in the first/second experiment.

Figure 7a,b exhibits the convergence processes of $\phi_{U}$ of the two fine alignment algorithms in the first and second experiments, respectively. These curves were obtained by off-line simulations. The starting points of Figure 7a,b were 4451.4 s and 5975.7 s, respectively. Due to the fast convergence of $\phi_{E}$ and $\phi_{N}$, there was little difference between the traditional methods and ESO methods. Hence, only the curve of $\phi_{U}$ was shown here. Figure 7a presents that the ESO method converged faster than the traditional method, and its alignment accuracy was slightly higher than the traditional method.

Figure 7b indicates that $\phi_{U}$ of ESO method was roughly stable after 5.5 min, and the misalignment angle (error) was small, while the traditional method needed longer convergence time and was easy to be disturbed by inaccurate speed measurement information. Table 4 shows the $\phi_{E}$, $\phi_{N}$, and $\phi_{U}$ at some critical moments. $\phi_{U}$ at some key time points have been shown in the above Figure 7a,b.

The results in Table 4 present that the accuracy and anti-interference ability of ESO algorithm are better than those of traditional algorithms. The misalignment angle of the heading angle of the proposed method is ≤3.3′ after 10 min of alignment. This demonstrates that an extension of alignment time of ESO will slightly improve the precision. In the calculation process, the consumptions of calculation time and hardware resources of this alignment method are statistically analyzed. The results show that the time and resource consumptions of the ESO method are far less (≪0.1) than those of the traditional KF method.

## 4. Conclusions

In this paper, ESO are applied in the algorithm of north-seeking, and an effective fast method is proposed. Three innovations of the proposed algorithm include ESO filter, TD filter, and the optimized parameters of these two filters. The advantages of this method are analyzed as follows: ① The experimental results show that the response characteristics of this method are outstanding. The results of first off-line simulation show that the misalignment angle of the heading angle of the proposed (traditional) method is ≤2.1′ (1.8′) after 5.5 (10) min of alignment. The results of second off-line simulation indicate that the misalignment angle of the heading angle of the proposed (traditional) method is ≤4.8′ (14.2′) after 5.5 (10) min of alignment. The simulations are based on the ship-running experimental data. The measurement precision of DVL is high (relatively low) in the first (second) experiment. Hence, the anti-interference ability of ESO can be investigated. These results demonstrate that the convergence speed, accuracy, and anti-interference ability of proposed ESO algorithm are better than those of traditional KF algorithm. ② The hardware resource consumption and time consumption of the ESO algorithm are far lower (≪0.1) than those of traditional KF algorithm.

This method can greatly shorten the azimuth alignment time, and its steady-state accuracy is very close to the theory limit. The anti-interference ability of the proposed algorithm is strong. The remarkable advantages show the effectiveness and practicability of ESO in the north-finding algorithm.

## Figures and Tables

**Figure 1 sensors-22-07547-f001:**
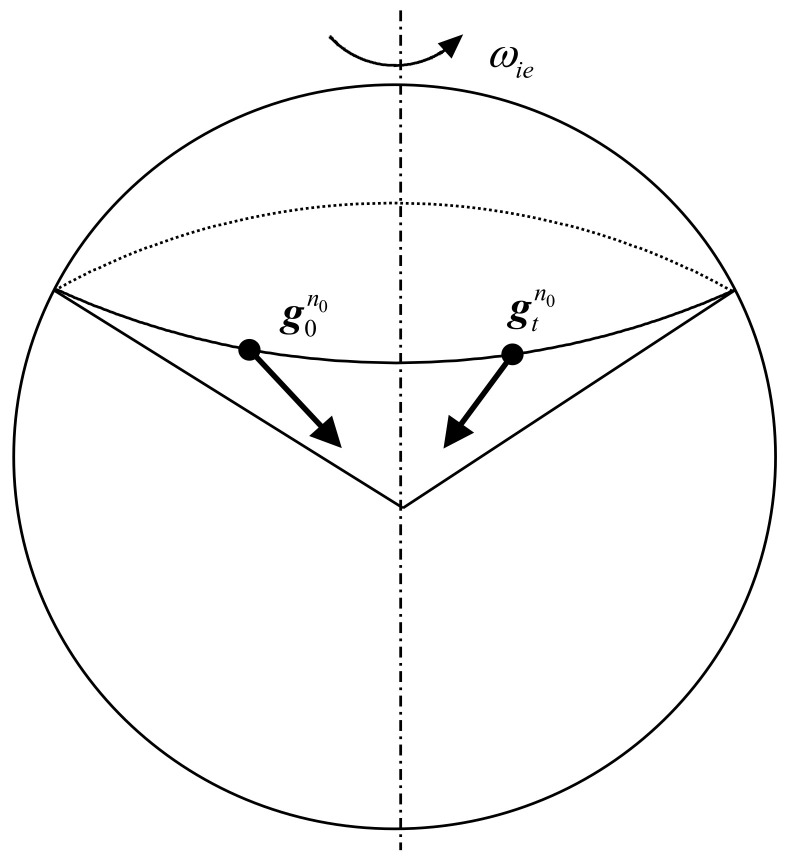
The schematic illustration of the cone constituted by the observed ***g***^*n*^^0^ in the inertial frame.

**Figure 2 sensors-22-07547-f002:**
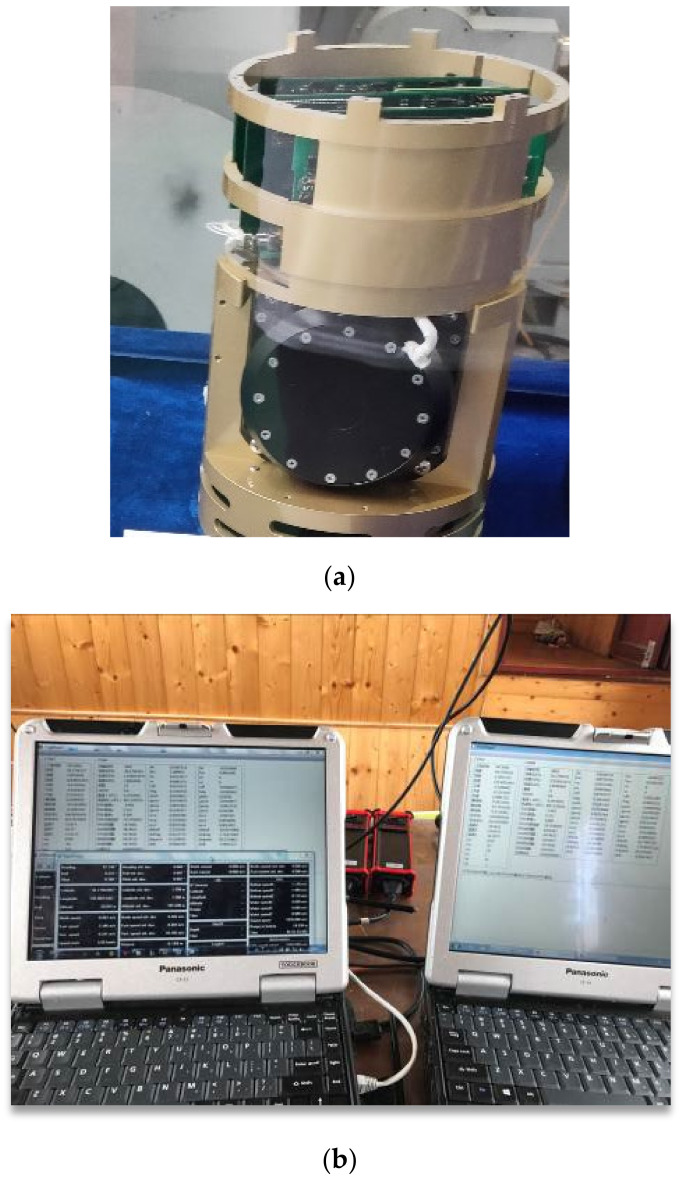
(**a**) The photograph of FOG north-seeker. (**b**) The photograph of the interfaces of test software of the north-seeker in the experiment.

**Figure 3 sensors-22-07547-f003:**
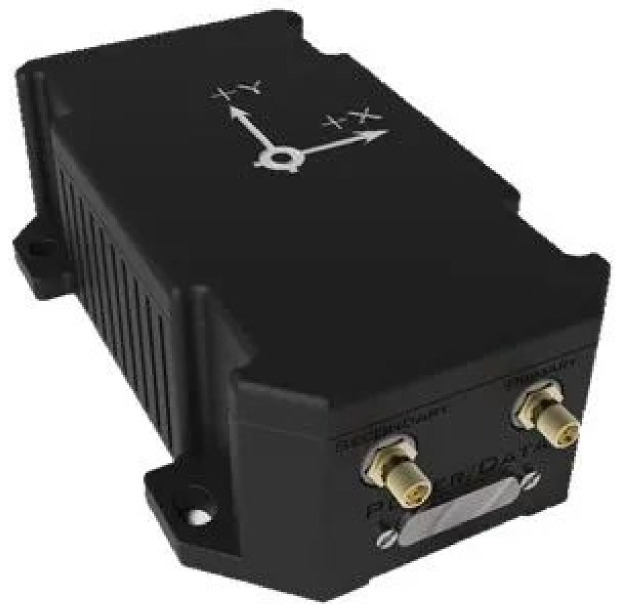
The photograph of RLG north-seeker.

**Figure 4 sensors-22-07547-f004:**
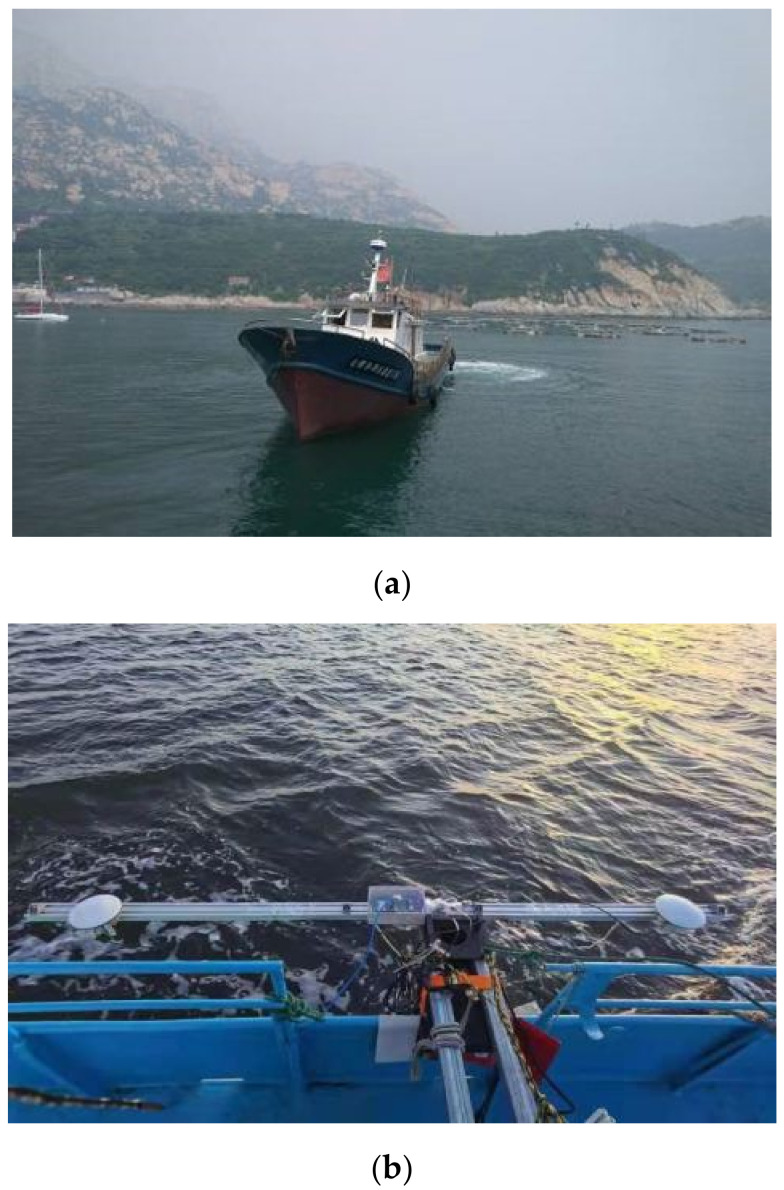
(**a**) The photograph of the experimental ship. (**b**) The photograph of the GNSS installed at the stern. (**c**) The photograph of the electronic box of DVL. (**d**) The photograph of the transducer (including emitter and receiver) of DVL.

**Figure 5 sensors-22-07547-f005:**
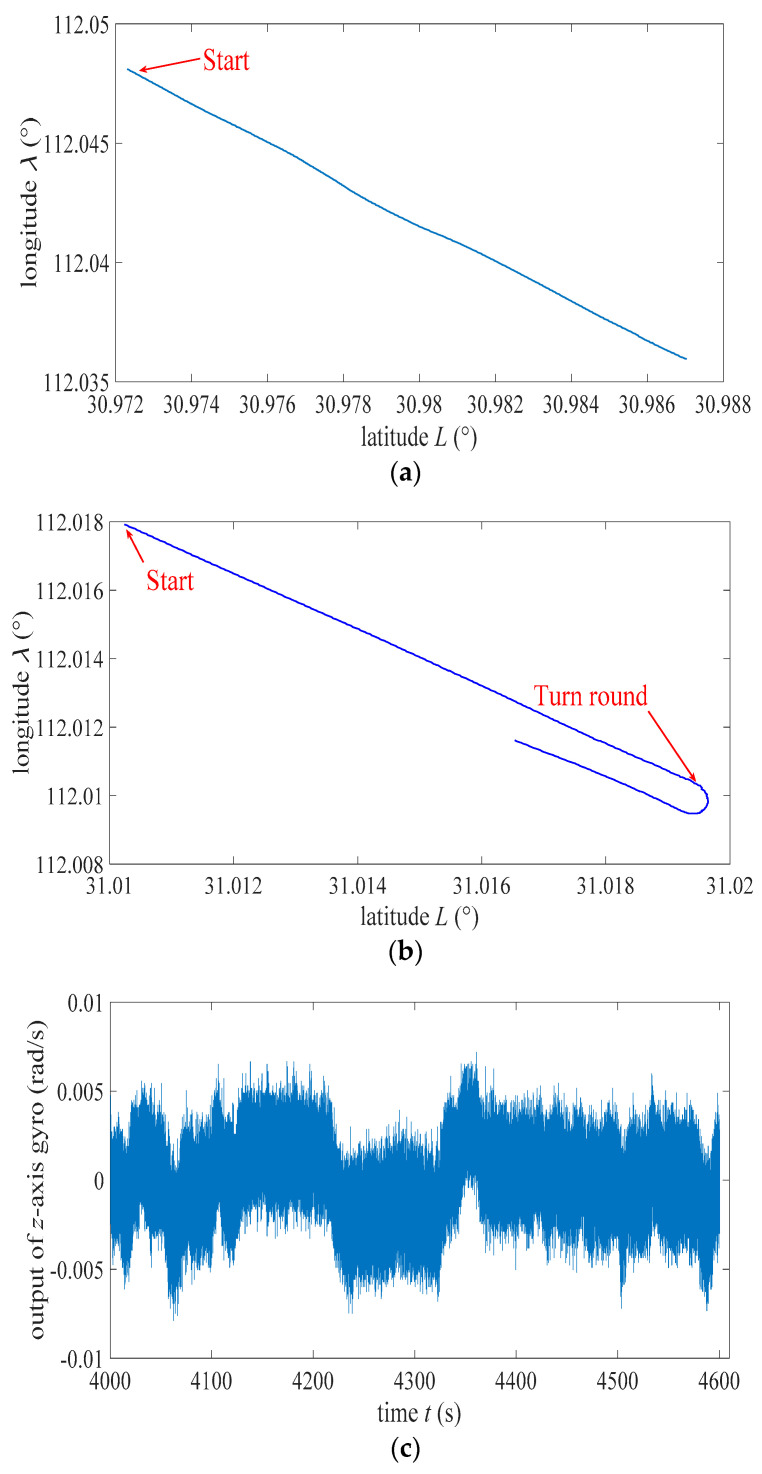
The diagrams of sailing trajectories based on the latitude and longitude coordinate of the ship in (**a**) first north-seeking experiment and (**b**) second north-seeking experiment. The measured data from *z*-axis gyro (**c**) and *z*-axis accelerometer (**d**) in first north-seeking experiment. The measured data from *z*-axis gyro (**e**) and *z*-axis accelerometer (**f**) in second north-seeking experiment.

**Figure 6 sensors-22-07547-f006:**
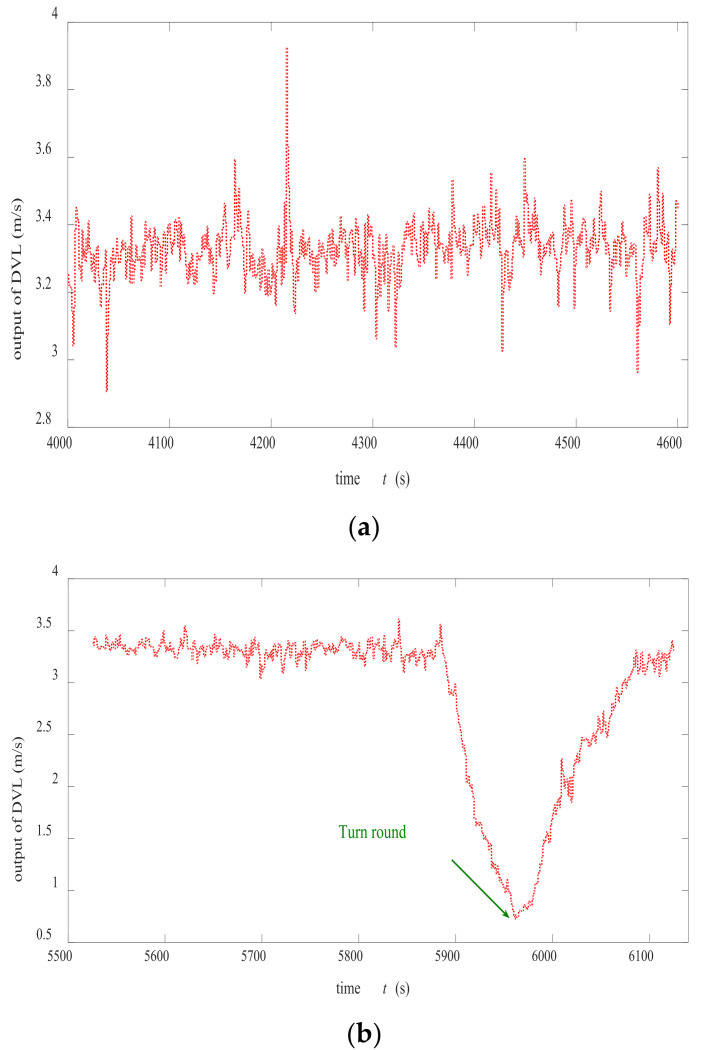
The diagram of the velocity measured by DVL during (**a**) first north-seeking experiment and (**b**) second north-seeking experiment.

**Figure 7 sensors-22-07547-f007:**
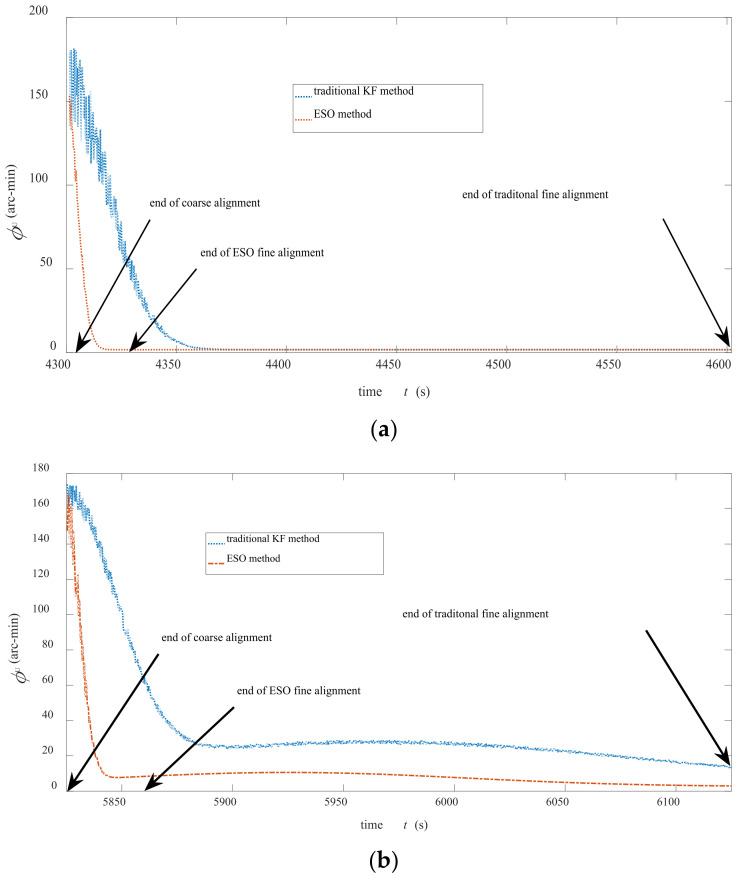
The results of misalignment angles in (**a**) first north-seeking experiment and (**b**) second north-seeking experiment.

**Table 1 sensors-22-07547-t001:** Errors of the inertial sensors in FOG north-seeker.

	Gyro Bias (°/h)		Accelerometer Bias (μg)	
	Constant	Random (White Noise)	Constant	Random (White Noise)
x-Axis	0.021	0.011	21	12
y-Axis	0.019	0.01	22	13
z-Axis	0.023	0.013	19	8

**Table 2 sensors-22-07547-t002:** $\phi_{N}$ and $\phi_{U}$ of the static experimental results.

	Static Experiment—Traditional KF Method	Static Experiment—Proposed ESO Method
$\phi_{E}$	5 min(coarse) + 0.5 min(fine): 0.4′ 5 min(coarse) + 5 min(fine): 0.08′	5 min(coarse) + 0.5 min(fine): 0.1′ 5 min(coarse) + 5 min(fine): 0.09′
$\phi_{N}$	5 min(coarse) + 0.5 min(fine): 0.7′ 5 min(coarse) + 5 min(fine): 0.2′	5 min(coarse) + 0.5 min(fine): 0.2′ 5 min(coarse) + 5 min(fine): 0.2′
$\phi_{U}$	5 min(coarse) + 0.5 min(fine): 19.2′ 5 min(coarse) + 5 min(fine): 0.6′	5 min(coarse) + 0.5 min(fine): 0.8′ 5 min(coarse) + 5 min(fine): 0.8′

**Table 3 sensors-22-07547-t003:** Errors of the inertial sensors in RLG north-seeker.

	Gyro Bias (°/h)		Accelerometer Bias (μg)	
	Constant	Random (White Noise)	Constant	Random (White Noise)
x-Axis	0.0011	0.0004	9	4
y-Axis	0.0007	0.0008	10	7
z-Axis	0.001	0.0007	10	5

**Table 4 sensors-22-07547-t004:** $\phi_{N}$ and $\phi_{U}$ of the two experimental results.

	First Experiment—Traditional KF Method	First Experiment—Proposed ESO Method	Second Experiment—Traditional KF Method	Second Experiment—Proposed ESO Method
$\phi_{E}$	5 min(coarse) + 0.5 min(fine): −2.5′ 5 min(coarse) + 5 min(fine): −0.1′	5 min(coarse) + 0.5 min(fine): −0.2′ 5 min(coarse) + 5 min(fine): −0.2′	5 min(coarse) + 0.5 min(fine): 0.4′ 5 min(coarse) + 5 min(fine): 0.5′	5 min(coarse) + 0.5 min(fine): 0.3′ 5 min(coarse) + 5 min(fine): 0. 3′
$\phi_{N}$	5 min(coarse) + 0.5 min(fine): 1.9′ 5 min(coarse) + 5 min(fine): 0.1′	5 min(coarse) + 0.5 min(fine): 0.2′ 5 min(coarse) + 5 min(fine): 0.2′	5 min(coarse) + 0.5 min(fine): −0.9′ 5 min(coarse) + 5 min(fine): −0.3′	5 min(coarse) + 0.5 min(fine): −0.6′ 5 min(coarse) + 5 min(fine): −0.5′
$\phi_{U}$	5 min(coarse) + 0.5 min(fine): 26.7′ 5 min(coarse) + 5 min(fine): 1.8′	5 min(coarse) + 0.5 min(fine): 2.1′ 5 min(coarse) + 5 min(fine): 2.1′	5 min(coarse) + 0.5 min(fine): 63.1′ 5 min(coarse) + 5 min(fine): 14.2′	5 min(coarse) + 0.5 min(fine): 4.8′ 5 min(coarse) + 5 min(fine): 3.3′

## Data Availability

Not applicable.

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
