# Peer review of "A Fast North-Finding Algorithm on the Moving Pedestal Based on the Technology of Extended State Observer (ESO)"

_sensors, 2022, doi:10.3390/s22197547_

Round 1

Reviewer 1 Report

The study is complete and the experiment is appropriate, but I believe more analysis is needed to demonstrate the advantage of the newly proposed method versus the state-of-the-art. Detailed comments below:

1. The authors claims that the new algorithm performs better than KF-based algorithm in terms of convergence time and accuracy. However, in (20), the authors increased IMU noise level by 50X. Of course it will make the KF less accurate and converge slower, so the comparison is not fair. The authors must address this point to make their statement solid.

2. The authors claim that Delta_E, Delta_N, and eps_E are unobservable. Why it's the case? In line 258, the authors set eps_E to be zero because of its unobservability. Why the authors can do it?

3. I believe more introduction is needed for ESO algorithm. For example, where do (30), (33), and (35) come from? What is the theoretical background of such equations?

4. Fig. 5 is very confusing, and not much information is conveyed.

Author Response

  1. The coefficient 50 was replaced by 1, and the static experiment and dynamic off-line numerical simulation were implemented or repeated again. The revised content are at lines 277, 279-281, and they are shown by table 2, table 4, and figures 7 (a) and 7 (b). The fine alignment of heading angle can be generally divided into two stages: rapid reduction (the first stage) and slow reduction (the second stage), as shown in Figs. 7 (a) and 7 (b). The coefficient of 50 is to increase the change step of heading angle during KF estimation, so the first stage will be accelerated. However, the accuracy of the second stage will be reduced. As suggested by the reviewer 1, coefficient 1 can ensure excellent performance of KF. Therefore, the comparison between ESO and KF model with coefficient of 1 is fair.

  1. The revised contents are at lines 303-309,317-320,322-327.

  1. The revised content is at lines 341-346.

  1. The revised content is shown at lines 471-475.

Reviewer 2 Report

1. I believe that there already exists a lot of related research work that the author should mention and cite in its introduction section.

2. The author should provide the organization of the article in the introduction section so that the readers should understand the workflow easily.

3. Figures quality need to be enhanced to better understanding.

4. In the abstract section, I would suggest that the author should provide to the point and quantitative advantages of the proposed method.

5. The literature review is poor in this paper. you must review all significant similar works that have been done. Also, review some of the good recent works that have been done in this area and are more similar to your paper. For each work first, explain the problem that has been addressed in that work. Then explain the aided to deal with that problem. After that, compare that work with your work and conclude the difference and the benefit of your work with that. The article can be further enhanced by connecting with some existing literatures. For example, The article can be further enhanced by connecting and comparing the undergoing work with some existing literatures. For example, 10.3390/agriculture12060793; 10.1109/JSTARS.2021.3059451 ï¼›10.1016/j.asoc.2022.109419   and so on.

6. Please highlight your contributions in introduction.

7. Please compare the pros and cons of existing solutions.

8. More equations are necessary to explain the proposed method.

9.  At Line 293, how to set  these values?

10.   At Line 407, add the sections of the “Institutional Review Board Statement”, “Informed Consent Statement”, “Data Availability Statement”.

Author Response

  1. The revised contents are at lines 86-126, 590-628.

  1. The revised description is at lines 137-142.

  1. The qualities and sizes of all figures have been improved.

  1. The revised content is at lines 26-31.

  1. The revised content is at lines 18-126.

  1. The revised content is at lines 127-136.

  1. The revised content is at lines 18-136.

  1. The revised content is at lines 360-384.

  1. The revised content is at lines 390-392.

  1. The revised content is at lines 518-523.

Round 2

Reviewer 1 Report

All issues addressed before have been solved.

Author Response

Responses to the Reviewer #2:

  1. The revised contents are at lines 170-180, 195-202.

  1. The revised description is at lines 59-70,95-120,149-155,160-166.

  1. The improved contents are at lines 591-593,601-604.

  1. The revised content is at lines 478-483.

  1. The added contents are on pages 5-10.

  1. The revised contents are at lines 59-70,95-120,149-155,160-166,621-727.

  1. The improved content is at lines 461-470.

Reviewer 2 Report

According to the revised paper, I have appreciated the deep revision of the contents and the present form of this manuscript. There is little content, which need be revised according to the comment of reviewer in order to meet the requirements of publish. A number of concerns listed as follows:

(1) Your contributions in introduction are not clear. Please further highlight  it.

(2)To explore Comparative results with existing approaches/methods relating to the proposed work. The method/approach in the context of the proposed work should be written in detail.

(3) Conclusion should be more carefully rewritten, summarizing what has been learned and why it is interesting and useful.

(4) The novel is not clear in Section 2, please highlight it.

(5) In the expressions(1)~(20), the meanings of the variables should be provided.

(6) The references in this paper mainly focus on those 10 years ago, which are too old-fashioned. Therefore, it is suggested that the author fuerther modify the references and cite the references in recent years as much as possible. Some suggested recent literatures should add in the revised paper.

(7) How to determine these parameters? The author should give a detailed explanation.

Author Response

(The authors gave the same response as above.)
